# Air Pollution Dispersion Modelling in Urban Environment Using CFD: A Systematic Review

**Mariya Pantusheva [1]**, **Radostin Mitkov [1]**, **Petar O. Hristov [1,2]** and **Dessislava Petrova-Antonova [1,*]**

[1]  GATE Institute, Sofia University "St. Kliment Ohridski", 1113 Sofia, Bulgaria
[2]  Institute for Risk and Uncertainty, School of Engineering, University of Liverpool, Liverpool L69 7ZF, UK
[*]  Correspondence: d.petrova@fmi.uni-sofia.bg

**Abstract:** Air pollution is a global problem, which needs to be understood and controlled to ensure a healthy environment and inform sustainable development. Urban areas have been established as one of the main contributors to air pollution, and, as such, urban air quality is the subject of an increasing volume of research. One of the principal means of studying air pollution dispersion is to use computational fluid dynamics (CFD) models. Subject to careful verification and validation, these models allow for analysts to predict air flow and pollution concentration for various urban morphologies under different environmental conditions. This article presents a detailed review of the use of CFD to model air pollution dispersion in an urban environment over the last decade. The review extracts and summarises information from nearly 90 pieces of published research, categorising it according to over 190 modelling features, which are thematically systemised into 7 groups. The findings from across the field are critically compared to available urban air pollution modelling guidelines and standards. Among the various quantitative trends and statistics from the review, two key findings stand out. The first is that, despite the existence of best practice guidelines for pollution dispersion modelling, anywhere between 12% and 34% of the papers do not specify one or more aspects of the utilised models, which are required to reproduce the study. The second is that none of the articles perform verification and validation according to accepted standards. The results of this review can, therefore, be used by practitioners in the field of pollution dispersion modelling to understand the general trends in current research and to identify open problems to be addressed in the future.

**Keywords:** systematic review; urban air pollution; pollutant dispersion; CFD; guidelines; verification and validation

## 1. Introduction

### 1.1. Why Air Pollution Is a Hot Topic

According to many organisations, air pollution is considered to be the largest environmental health risk, with a significant impact on the everyday life of citizens, particularly in urban areas. Although the concentrations of key air pollutants in the ambient air in Europe have decreased over the past two decades, air quality continues to be poor, causing a wide range of diseases [1]. The latest World Health Organization (WHO) fact sheet dedicated to ambient pollution shows that in 2019, 99% of the world population was living in places where the levels in the WHO air quality guidelines were not met [2]. Exposure to air pollution can lead to cardio-vascular, respiratory and pulmonary diseases, lung, bronchus and trachea cancer, and asthma. Furthermore, it has a negative impact on cognitive performance in both children and adults, and can relate to conditions such as dementia, Alzheimer's, and Parkinson's diseases [3,4]. According to the WHO, it causes millions of deaths and disabilities worldwide [5].

Air pollution directly impacts climate change by altering the energy balance between the atmosphere and the surface of the Earth. The complex interactions in the atmosphere cause both thermal and chemical composition changes [6]. Climate change increases the production of allergenic air pollutants such as pollen and mould, which can be hazardous to health [7]. An important economic consequence of health deterioration and climate change is the decreased labour productivity and increased medical costs. The Organization for Economic Cooperation and Development (OECD) has reported that the annual number of lost working days is projected to reach 3.7 billion at the global level by 2060 [8]. Various other negative consequences are reported to be due to air pollution, including reductions in visibility, damage to materials, and ecological damage such as acid rains and reduced agricultural productivity [9].

Due to the present challenges, the topic of climate-neutral, sustainable smart cities is becoming more and more relevant each day. It is part of the Global Goals, [10], and the EU Missions defined under Horizon Europe research and innovation programme, [11]. Globally, much effort is focused on the transition to more resilient, greener, healthier, and more inclusive urban environment based on scientifically proven solutions, and the utilization of technology and innovations, i.e., smart cities. Air quality can be considered an important smart city indicator, [12]. With the increasingly rapid global urbanisation, reducing air pollution has become a significant challenge, leading to a growing number of studies dedicated to understanding the problem and mitigating the related risks. Some of the efforts for the improvement of air quality include: (1) the use of mechanical factors to reduce air pollution, such as natural, forced and traffic-induced convection; (2) exploring dependencies between air pollution and urban design [13,14]. Findings suggest that the urban landscape, building geometries, and various mechanical means are important measures to control air pollution. In this regard, some authors have introduced the definition of different urban and ventilation indices, describing city breathability and supporting a sustainable building design [15,16]. However, such parameters are still not widely utilized in urban planning and have not yet been included in a legislative framework.

### 1.2. Pollution Dispersion Modelling

As with an increasing number of complex problems in science and engineering, computational models are being applied to understand and predict air pollution dispersion [17–21]. These models support urban planners to find solutions to improve air quality and thus safeguard human health, decrease environmental damage and reduce economic losses. Choosing an appropriate and reasonably efficient modelling framework is essential to appropriately handle the problem and successfully model air pollution dispersion [19].

Atmospheric computer models are typically divided into three main categories based on their spatial scope. According to Orlanski's atmospheric scale [22], they can be defined as microscale-level models (scope less than 2 km), mesoscale (2 km–2000 km), and macroscale or synoptic scale models (covering an area larger than 2000 km). For microscale pollutant dispersion, however, Blocken et al. [23] suggest a 10 km horizontal length scale. The main interest of this study is the meteorological and pollution dispersion microscale level from the perspective of the urban environment. At this level, computational fluid dynamics (CFD) models have been utilised to study both meteorological and air pollution dispersion phenomena, since they provide insight into different flow variables throughout the calculation domain and avoid similarity requirements, which often affect wind tunnel tests [24]. CFD simulations may use boundary conditions from larger scale models as an input, parameterise part of the turbulence, but explicitly model the relevant urban obstacles, thermals, building wakes, convection, and large-scale turbulence [25]. They provide detailed spatial and temporal predictions, but require a significant computational time and resources in comparison, for example, with the Gaussian modelling approach [26]. The accuracy of CFD models is very sensitive to the selected parameters and conditions [27] and, as a result, best practice guidelines have been compiled to help choose these appropriately.

*1.3. Related Work*

Several literature reviews have been published in the field of CFD modelling of air pollution dispersion, which attempt to determine the trends in the field at the time of the review, what parameters are important to accurately model pollution dispersion and how the model can be made more efficient. Several of those that were important in designing the current review are discussed below.

A systematic review of the effects of different factors that influence the pollutant dispersion and quantification of the potential to reduce pollution by changing these parameters was conducted by Li et al. [13]. There, it was found that morphological parameters such as the heterogeneity of the atmospheric boundary layer (ABL), urban density, building openings, and others, significantly affect air pollution dispersion. Furthermore, the inflow wind condition, vehicular motion, and thermal effects are found to be mechanical measures that have similar levels of influence.

An overview of the main characteristics of the CFD simulation of microscale pollutant dispersion in the built environment is provided in [23]. The article presents an overview of papers that were previously published in the journal "Building and Environment". The explored studies are compared based on their configuration, e.g., two-building, street canyon, and idealised city model, as well as turbulence model, assessment method, validation and sensitivity analysis.

Around the same time, Di Sabatino et al. [17] reviewed the literature for the effects of vegetation, traffic-produced turbulence and thermal phenomena on a range of different urban morphologies at two different scales. For instance, urban canyons and street intersections were investigated at the single-building and street scale (microscale), while more realistic geometries and arrays of regular buildings were considered to reside at the district level. Interestingly, the authors mention street canyons in the district scale as well, demonstrating the wide boundary between scales. In accordance with previous work, the review suggests that many parameters affect atmospheric and pollution dispersion dynamics, including both natural (wind and thermal effects) and human-induced (built environment and vehicular motion) mechanisms. The review also discusses computationally cheaper alternatives to CFD for complex urban geometries. This discussion is also supported by the mathematical model review provided in [19], where the authors examine different issues related to performance, including parallel and graphical-processing-unit-based computing.

In contrast to [17], Tominaga and Stathopoulus [28] reviewed a number of studies on near-field pollutant dispersion around buildings using CFD. They defined 'near-field' as the flow region immediately surrounding an obstacle, splitting the configurations used in reviewed studies into four categories: an isolated building, a single street canyon, building arrays, and building complexes. The review identifies key features of models for near-field pollutant dispersion around buildings from previous studies.The review concludes that to adequately model near-field phenomena, the model must reside in a three-dimensional domain, with the gain in accuracy overturning the decrease in efficiency. Special attention was also paid to the importance of carefully choosing turbulence models and parameters, as well as boundary conditions.

Another review, focused on near-field pollution dispersion, is given by Lateb et al. [20]. The authors chose to investigate issues related to the modelling of wind flow around buildings, as they deem this to be the driving factor behind accurate pollution dispersion models. After reviewing the relevant literature, the authors provide a number of useful working definitions for various concepts, such as near-field and far-field scales being subdivisions of the microscale level, components, and characteristics of the ABL, among others. The review also points to the need and importance of verification and validation (V&V) efforts, which are integral to pollution dispersion modelling.

Continuing on the topic of V&V, Herring and Huq [26], reviewed the most widely adopted methods to assess the predictive performance of pollution dispersion models. The authors recognise the importance of critically assessing models if their output is to be used in decision-making. The authors determine that most of the reviewed studies use a single

source of inspiration when it comes to V&V, and one that has some obvious issues. Matters related to V&V are discussed in more detail in Section 3.9 of the current paper.

More recently, Zhang et al. [21] produced a review that pays special attention to how the flow structure inside street canyons is affected by objects and processes external to the model itself. Some of the investigated objects include noise barriers and vegetation, while solar radiation and wall heating exemplify the considered processes. The authors determine that most of the external objects and processes affect the pollution dispersion weakly compared to geometric factors, such as canyon height difference and inflow conditions.

Recommendations regarding the application of CFD to predict the pedestrian wind environment around buildings have been proposed by the European Cooperation in the field of Scientific and Technical Research (COST) group [29]. They are focused on microscale obstacle-accommodating meteorological models, without considering dispersion modelling. The recommendations are based on a review of the currently available guidelines in previous research. The working group of the Architectural Institute of Japan (AIJ) has taken a different approach, and has proposed guidelines for the practical application of CFD to solve the same problem based on a cross-comparison of the results from experiments and tests [30]. CFD predictions, wind tunnel tests, and field measurements for a number of test cases have been conducted and analysed to explore the influence of different kinds of conditions in the urban environment on the flow field. The existing CFD best-practice guidelines are complemented by the focused research efforts of other authors [24,31,32]. Their recommendations cover the whole process of modelling and are related to the development of the computational domain and the computational grid, the determination of roughness parameters, the setting of inlet boundary conditions and convergence criteria, the selection of higher-order discretisation schemes, the testing of horizontal homogeneity, and grid convergence analyses, validation and reporting.

The Verein Deutscher Ingenieure (VDI) is one of the largest associations of engineers and natural scientists in Western Europe. They have developed a database with over 2100 valid standards, which aims to provide practice-oriented technical guidelines in many engineering fields [33]. One of these standards is related to environmental meteorology and air pollution. The VDI 3783 series (parts 1 to 21) refer to the flow and dispersion processes in the atmospheric boundary layer. Parts 1, 2, 4, 8, and 13 [34–38] provide general information on pollutant dispersion from different sources, and guidelines for assessment and modelling. Parts 9, 10, and 12 [39–41] specifically refer to microscale modelling requirements for wind flow and pollutant dispersion in the ABL, considering the orography and environmental conditions.

The current study presents an overview of the air pollution modelling techniques in an urban environment based on 74 publications from the period 2012–2022. In addition, where applicable, existing best practice guidelines are summarized and compared to the methods used in the examined articles. A key contribution of this paper is its methodical and structured approach to summarising important aspects of the published research on urban air pollution modelling. To that end, the paper provides a comprehensive and detailed description of the CFD model creation process, including relevant settings and options available to the analyst. The paper aims to assist engineers and researchers working on urban simulations and enable them to make informed choices when developing their analysis models.

The paper, as seen in Figure 1, is organised as follows. The research methodology is described in Section 2. The selection of different model features for the review and the review process itself are discussed in Section 3. The process of extracting and processing information is described in Section 3.1. The overview begins with information on general model features in Section 3.2. This is followed by a discussion of the technical aspects of the modelling process, such as the selection of geometry dimensionality in Section 3.3, and computational domain and mesh settings in Sections 3.4 and 3.5, respectively. The physics and boundary conditions are presented in Sections 3.6 and 3.7. The final component of the modelling process, the solution setup, is described in Section 3.8. The review of the

selected studies concludes in Section 3.9 with a note on verification, validation, uncertainty quantification, and predictive capability. Final remarks are presented in Section 4.

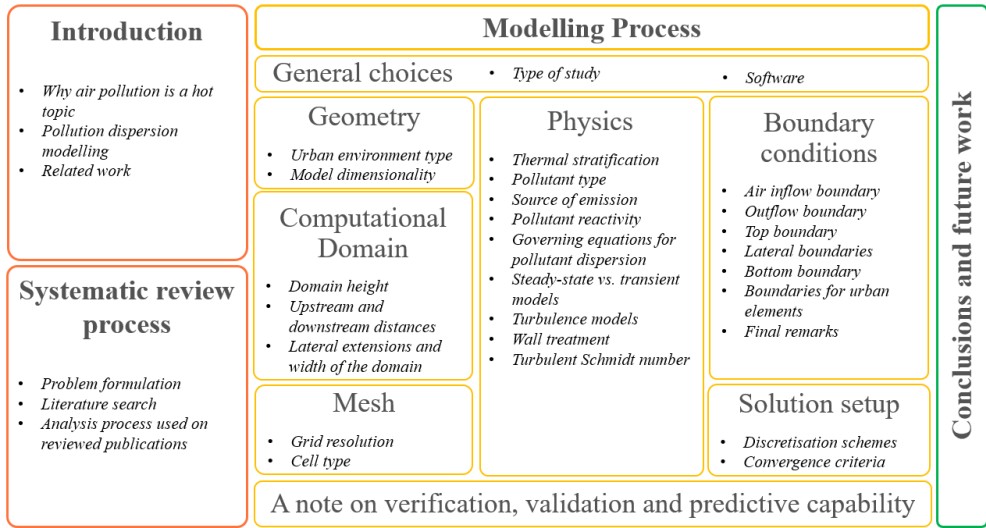

**Figure 1.** Structure of the paper.

## 2. Materials and Methods

The research methodology follows the guidelines for systematic literature reviews proposed by Kitchenham et al. [42], including three phases: (1) planning the review, (2) conducting the review and (3) reporting the review.

### 2.1. Problem Formulation

The specificity of the research field, the growing popularity of CFD, and the increasing number of publications on the topic have emphasized the lack of unified guidelines for the use of CFD for air pollution modelling. To provide a coherent examination of the state-of-the-art in the field, this systematic review formulated the following research questions:

1. What is the scope of the current CFD models of air pollution dispersion in terms of geometry? (RQ1 discussed in Section 3.3)
2. What are the most commonly used mesh types and their characteristics? (RQ2 discussed in Section 3.5)
3. What are the most commonly used turbulence models and their settings? (RQ3 discussed in Section 3.6)
4. What are the most commonly used domain parameters, boundary conditions and settings for the solution setup? (RQ4 discussed in Sections 3.4, 3.7 and 3.8)
5. What validation and verification measures are recommended for CFD models of air pollution dispersion? (RQ5 discussed in Section 3.9).

### 2.2. Literature Search

The online databases Scopus and Science Direct were selected as appropriate information sources for the purposes of this study. In addition, a literature search was performed in the scientific search engine Google Scholar. Based on the research questions, the following keywords were used as initial criteria to find relevant studies:

*"air pollution" AND "pollutant dispersion" AND "urban" AND "CFD simulation".*

The search was performed in such a way that all the keywords were present in the full-text or metadata of the papers. Furthermore, a search filter was applied, limiting the publications to those published in the last ten years (2012–2022). Due to the large number of papers that were obtained, two more search terms were added to additionally focus the search on papers concerning only urban CFD applications, as follows:

*"building" AND "mesh"*.

This narrowed down the relevant papers for review. The number of articles that matched the initial and narrow search is presented in Table 1.

**Table 1.** Initial and narrow search of relevant studies.

| No | Database | | Initial Search | Narrow Search |
|----|----------|---|----------------|---------------|
| 1 | Scopus | | 715 | 103 |
| 2 | Science Direct | | 230 | 147 |
| 3 | Google Scholar | | 1090 | 668 |
| | | **Total** | 2035 | 918 |

A method referred to as a 'bibliographic search', 'citation tracking', 'snowballing', or 'pearl growing' was applied as a complimentary technique to identify more papers relevant to the current study [43,44]. As a result, ten additional publications were included in the literature review. A manual search in sources known to the research team and a general search on the Internet was also performed to deepen the study. Some works published before 2012 were also included since they were considered very relevant to the scope of the literature review.

The papers identified in the narrow search were further filtered based on a screening of their title and abstract. Due to the high number of publications identified through Google Scholar, only the first 150 papers (out of the identified 668), based on the "relevance" criterion provided by the web search engine, were selected for further processing. Duplicate records from the different databases were removed. The general process of the literature search is presented in Figure 2.

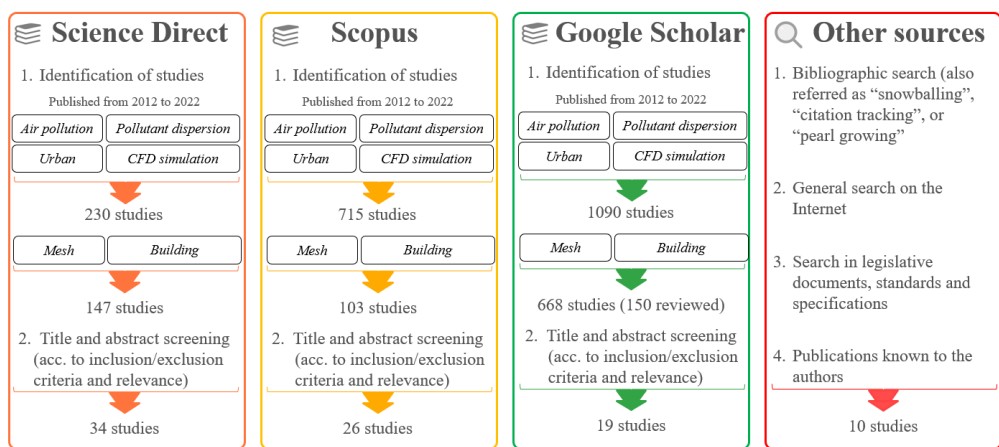

**Figure 2.** Literature search process.

The literature filtering was performed according to the inclusion and exclusion criteria and the relevance of the papers to the scope of this study. The inclusion criteria were defined as follows:

1. Articles that are peer-reviewed;
2. Articles that were published after 2012;
3. Articles that provide specific insights on the CFD setup of air pollution dispersion modelling in an urban environment.

The exclusion criteria were defined as follows:

1. Articles that are primarily focused on the measures to decrease air pollution rather than on the CFD simulation setup;
2. Articles that are primarily focused on wind comfort or thermal comfort rather than air pollution dispersion;

3. Articles that are primarily focused on the ventilation, building configuration, balconies, roof type and sound barriers, rather than on the best approach for modelling pollution dispersion.

The results from the evaluation of the articles based on the above criteria are shown in Table 2.

**Table 2.** Total number of articles included in the literature review.

| No | Type of Search | | Number |
|----|----------------|---|--------|
| 1 | Scopus | | 26 |
| 2 | Science Direct | | 34 |
| 3 | Google Scholar | | 19 |
| 4 | Bibliographic search | | 6 |
| 5 | Manual search | | 2 |
| 6 | Other relevant work published before 2012 | | 2 |
| | | **Total** | 89 |

The final scope of the literature review was limited to 89 articles, 9 of which were review papers used for reference, 5 providing best practice guidelines and another one that constituted a standard for environmental meteorology and pollution dispersion modelling. The remaining 74 articles, which form the core of the current review, discuss different applications of CFD modelling for pollution dispersion.

## 3. Discussion

For this overview, the criteria for over 190 model features were identified and related information was extracted from each paper (if available). The criteria were divided into seven main groups, namely: general information, geometry (domain settings), mesh, physics, boundary conditions, solution setup, and V&V. The specifics and settings in each of these sections influence the CFD model results and reliability. A short summary of the selected key factors from each of the above categories is presented in the following text.

### 3.1. Analysis Process

The data presented for each parameter were obtained following the procedure described below:

1. Information from each paper for the investigated parameter is extracted, if available. This information can vary in terms of its representation. The domain dimensions, for example, can be expressed in terms of absolute metric units, building height, or another relevant measure. A domain dimension can also denote either the offset distance from the building to the domain boundaries, or the total distance between these boundaries.

2. The extracted information about all of the distinct representations is unified so that the information can be grouped and compared against each other.

3. In cases where a parameter adopts multiple values in a single publication (for instance, the distance between the top of the computational domain and the tallest building; see Section 3.4), only the least favourable (least conservative one) is included in the analysis.

4. Information is sorted and filtered, if necessary. For example, the parameter *domain width* is dependent on another parameter, which is the dimensionality of the model (2D or 3D). The *domain width*, defined as the distance between the built area and the lateral boundaries of the domain (not in the flow direction), does not exist in a two-dimensional domain. Therefore, papers that utilize 2D modelling need to be filtered and excluded from the analysis of the parameter *domain width*.

5. Finally, the information is grouped in a sensible and comprehensive manner, suitable for chart depiction and further analysis. The logic behind this selection is highly dependent on the investigated parameter. In the following paragraph, the *domain height*

parameter group selection is provided as an example. The best practice guidelines (BPG) [29,30] recommend at least $5H$ offset from the top of the building under investigation to the top boundary of the domain, i.e., at least $6H$ domain height. Doubling this recommendation (offset of $10H$ above the building of interest) would lead to $11H$ total domain height. Therefore, the selected groups are defined as follows:

Group 1: The domain height is less than the BPG recommendation (domain height $< 6H$).

Group 2: The domain height is in the range between the BPG recommendation and its doubled value ($6H \leq$ domain height $\leq 11H$).

Group 3: More conservative studies are included where the domain height is greater than the double the value of the BPG (domain height $> 11H$).

Group 4: Papers that do not explicitly state the domain height and only claim that they follow the BPG.

Group 5: The domain height is not specified or mentioned at all.

A summary of the analysis process for each parameter is presented in Figure 3.

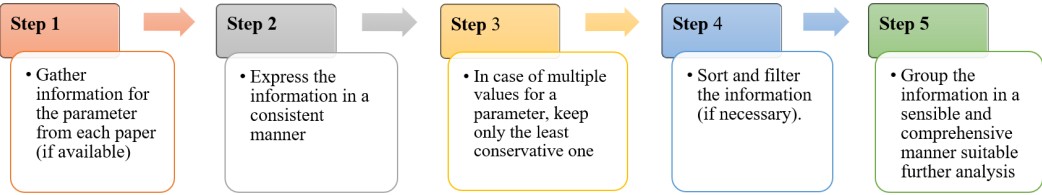

**Figure 3.** Parameter analysis process.

*3.2. General Choices*

In this section, some of the fundamental features of the publications, such as the type of study and the software used for computation, among others, are investigated.

### 3.2.1. Type of Study

A CFD study can serve one of two general purposes: it can be used to solve a real-life problem (industrial, urban planning, HVAC, etc.), representing real geometries and a real urban environment (referred to as an *applied* publication), or it might be there to question or enrich the existing knowledge in the particular science field to which it belongs (referred to as a *research* publication). The latter could be performed on real city geometries, although it is more common to use simplifications in terms of building shapes and layouts. Some publications could investigate both generic and city scenarios, and are thus suitable for both *applied* and *research* purposes. For the 74 reviewed candidate papers covering the air pollution dispersion modelling in urban environments, the analysis shows that most of the publications released in the last decade are focused on *research* activities (65%). More detailed information on the publication type distribution is presented on Figure 4.

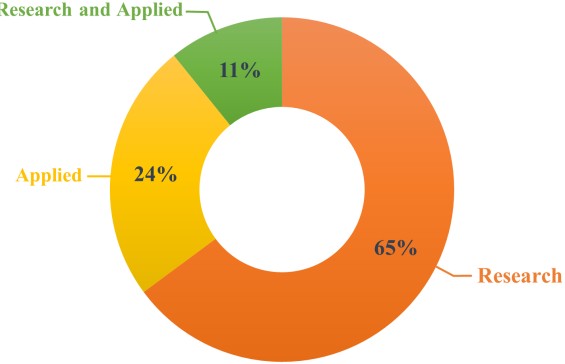

**Figure 4.** Publication type.

### 3.2.2. Software

The authors believe that the choice of suitable computational fluid dynamics software would be of interest to the researchers and engineers working in the field. An overview of the software packages used in the reviewed publications is presented in Figure 5. It should be noted, however, that such a breakdown is not necessarily an objective representation of the capabilities or deficiencies of different tools. There may be non-technical reasons behind the choice of a particular computing environment, such as availability, resource limitations and user base loyalty, which are beyond the scope of this work.

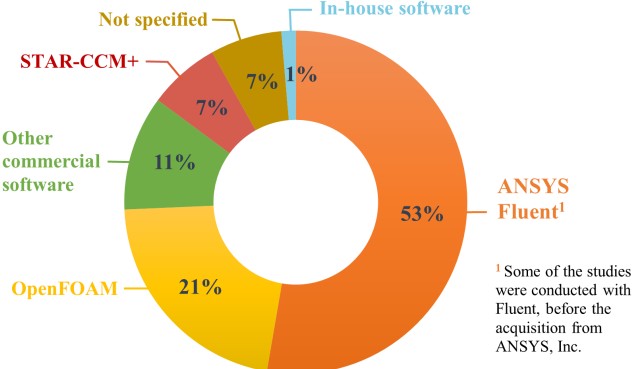

**Figure 5.** Software usage in publications.

The software usage analysis can be further detailed considering the publication type; see Figure 6. The results mainly follow the same trends as the ones presented in Figure 5; ANSYS Fluent is the most commonly used software in both *research* and *applied* publications, followed by OpenFOAM. Star-CCM+ takes the third place in *applied* applications; however, it is not used for purely *research* purposes. For both types of studies, other commercial software is used in approximately 10% of the publications. The in-house developed code was only utilized for *research* publications. The software selection is important in the sense that it determines the set of calculation methods, physics, boundary conditions, and solution settings available to the analyst. A trend was observed: in 8% of the *research*, and 4% of the *applied* publications, the software used for the CFD modelling is not specified.

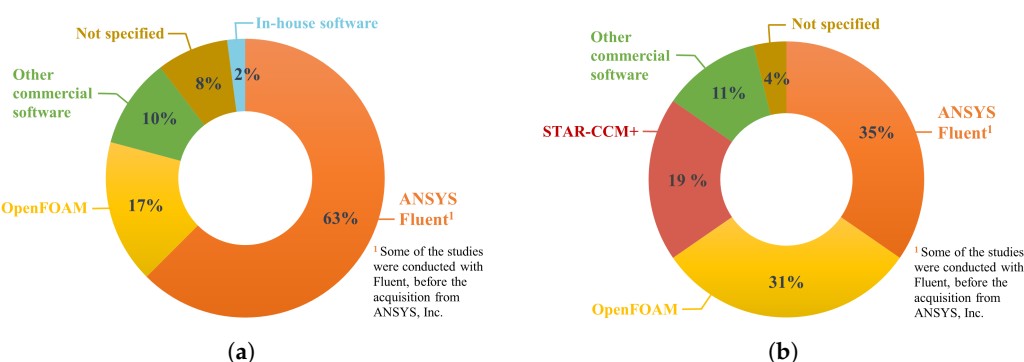

| (**a**) | (**b**) |

**Figure 6.** Software usage by publication type. (**a**) *Research* publications; (**b**) *Applied* publications.

### 3.3. Geometry

This section will discuss some high-level features of the model's geometry, which need to be selected prior to initiating the analysis process: namely, the model's dimensionality and the urban environment type. The selection of computational domain and boundary conditions will be covered in detail in Sections 3.4 and 3.7.

### 3.3.1. Urban Environment Type

The authors of this paper have collected information regarding the urban environment type used in the selected publications in the sense of whether the model includes, for

example, a single building, a street canyon, multiple building blocks, or a real city landscape. However, the descriptions in the reviewed publications vary widely, making it difficult to harmonise a representative classification of geometry. This precludes the quantitative assessment of results and their graphical representation. Nevertheless, the summary information is available upon request.

### 3.3.2. Model Dimensionality

In the presence of physical obstacles (such as those found in the urban environment), wind flow and turbulence exhibit complex three-dimensional (3D) behaviour. However, 3D simulations could be quite demanding in terms of computation resources and calculation time. In certain situations, two-dimensional (2D) simulations can be utilized to simplify the analysis. These cannot capture the complex 3D phenomena of the flow but may be adequate for use when the effects in one of the three directions can be neglected. To verify the latter, some authors used 2D analysis in their work, but validated the results with 3D simulations. Another possible option in CFD modelling is the so-called 2.5D model. This is a hybrid between a 2D and a 3D model—in essence, it is two-dimensional but with an artificially added third dimension. The latter is achieved using periodic boundary conditions and is most often applied to 3D geometries that are homogeneous in the span-wise direction (e.g., long street canyons).

The distribution between the different dimensionalities used in the reviewed papers is presented in Figure 7. The majority of authors, 77%, use 3D models in their analyses, 18% use 2D models, and the remaining 5% utilize either 2.5D, or both 2D and 3D models. One of the publications [45] presents an overview of the experimental and modelling studies published in the last two decades and states that previous studies show significant differences in the airflow and dispersion between 3D and 2D canyons.

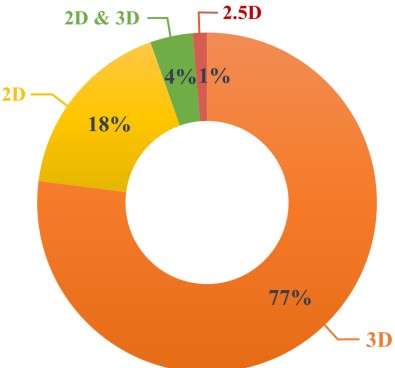

**Figure 7.** Model Dimensionality.

### 3.4. Computational Domain

The shape and size of the computational domain are crucial for the accurate representation and solution of external aerodynamics problems. The authors of 85% of the reviewed papers report that the chosen domain shape is a box (or a rectangle in the 2D case). This choice affects subsequent uses of the model for parametric wind studies.

Domain size can highly affect the accuracy of the simulation results. Insufficient offset dimensions from the investigated urban landscape to the domain boundaries can cause significant flow disturbances, leading to non-physical results. On the other hand, too-large domain dimensions can lead to an excessive number of mesh cells and unnecessarily increase the computational resources and time required for the simulations. Therefore, the best option is to achieve a balance between the two. The optimal domain dimensions, represented by their offsets from the investigated building(s) to the domain boundaries, have been a topic in two major projects that produced best practice guidelines in recent decades, [29,30], in some standards [39], and other research [32,46]. The general recommendations related to these research efforts are presented in the following sections.

### 3.4.1. Domain Height

The best practice recommendations from [29,30] regarding the *domain height* selection for CFD models of urban air pollution are summarized in Table 3.

**Table 3.** Best practice guidelines on domain height.

| Publication Title | Year | Single Building | Multiple Buildings | Wind Tunnel Experiment | Real Terrain with Building Surroundings |
|---|---|---|---|---|---|
| Best practice guidelines for the CFD simulation of flow in the urban environment, quality assurance and improvements in microscale meteorological models (COST 732) [29] | 2007 | Minimum $6H$ | minimum $6H$, where $H$ is the height of the *tallest* building | Minimum of (the wind tunnel's test section height; $6H$), where $H$ is the height of the *tallest* building | N/A |
| AIJ guidelines for practical applications of CFD to pedestrian wind environment around buildings [30] | 2008 | Minimum $6H$ | Minimum $6H$, where $H$ is the height of the *target* building | N/A | The height of the computational domain should be set to correspond to the boundary layer height determined by the terrain category of the surroundings [47] |

The standard VDI 3783 [39] also provides prescriptions for the domain dimensions. This states that, if the upper edge of the domain is too low, then this can result in a distortion of the flow fields. Therefore, the domain height should be chosen so that the *blockage ratio* is less than 10%. The *blockage ratio* is defined as the maximum built-up area in any plane perpendicular to the relevant flow direction related to the model's sectional plane area (*width* $\times$ *height* of the domain) in the same direction.

From the reviewed publications, there is one more paper that provides recommendations for the domain height, [46]. It states that a domain height of $7.5H$ is recommended to minimize the blockage effect. However, this analysis is based on an LES model of a street canyon and, therefore, cannot be categorized in the general sections presented in Table 3. The other reviewed publications chose a domain height for their analyses without giving domain size recommendations. An overview of the selected domain height used in the reviewed CFD studies is presented in Figure 8.

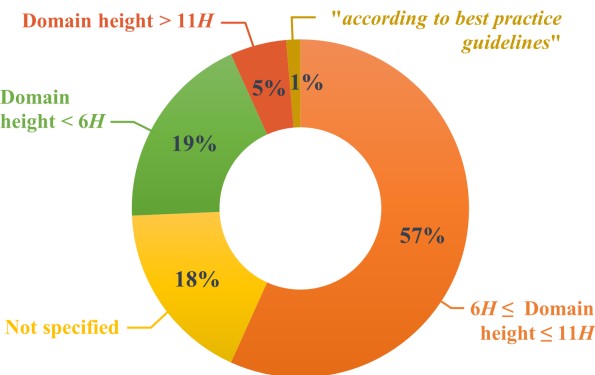

**Figure 8.** Domain height overview.

It should be noted that the majority of the publications (63%) follow the best practice guidelines for domain height and accept this dimension to be equal or greater than $6H$. Some of them even use much higher values: $12H$ [16], $17H$ [48], $25H$ [49], and $33.3H$ [50]. However, there is still a rather high percentage (19%) that do not comply with the existing recommendations. In these studies [51–53] the vertical extension of the domain is only

2*H*, and in [54,55], this is reported as 1.5*H* and 1.2*H*, respectively. Eighteen percent of the articles do not report on the domain height used in their analysis. Considering that this parameter can artificially accelerate the wind flow, the authors of this review strongly advise that the domain height is reported, the BPGs are followed, and every CFD model is tested for results' sensitivity with respect to the domain height.

### 3.4.2. Upstream and Downstream Distances

The *domain upstream distance* is defined here as the horizontal distance, measured from the domain inflow boundary to the closest building or other urban element under investigation. Similarly, the *domain downstream distance* denotes the distance behind the measured built area to the domain outflow boundary. The best practice recommendations from [29,30] regarding the domain dimensions in the flow direction are summarized in Table 4.

**Table 4.** Best practice guidelines on domain upstream and downstream distances.

| Publication Title | Year | Domain Upstream Distance, [*H*] | Domain Downstream Distance, [*H*] |
|---|---|---|---|
| Best practice guidelines for the CFD simulation of flow in the urban environment, quality assurance and improvements in microscale meteorological models (COST 732) [29] | 2007 | Minimum 5*H* if the approach profiles are well known. If the approach profiles are not available, a larger distance should be used to allow for realistic flow establishment | Minimum 15*H* to allow for flow redevelopment behind the wake region |
| AIJ guidelines for practical applications of CFD to pedestrian wind environment around buildings [30] | 2008 | Should be set to correspond to the upwind area covered by a smooth floor in the wind tunnel | Minimum 10*H* |

These guidelines are generally prescribed for single-building models. For an urban area with multiple buildings, Ref. [29] suggests that a smaller distance than 15*H* can be used for the downstream distance if the flow entering through the outflow domain boundary can be avoided, as this can negatively affect the convergence of the solution. VDI [39], on the other hand, does not provide any clear recommendations for minimum distances between the built area and the domain boundaries in the flow directions, but suggests that they are set based on a blockage ratio of no more than 10%.

In the reviewed papers, the majority of authors follow the best practice recommendations discussed above. The information concerning the upstream and downstream distances was extracted from the reviewed papers and processed based on the procedure explained in Section 3.1. The breakdown of the results is presented in Figures 9 and 10.

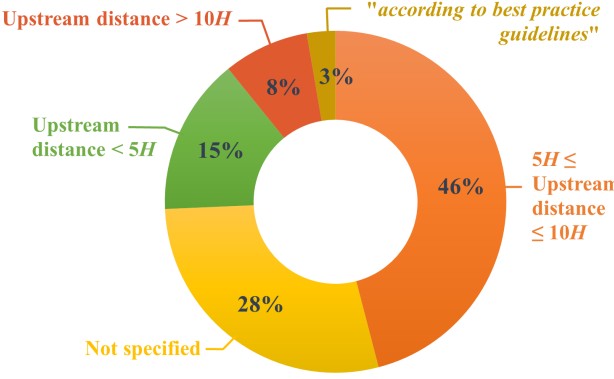

**Figure 9.** Upstream distance overview.

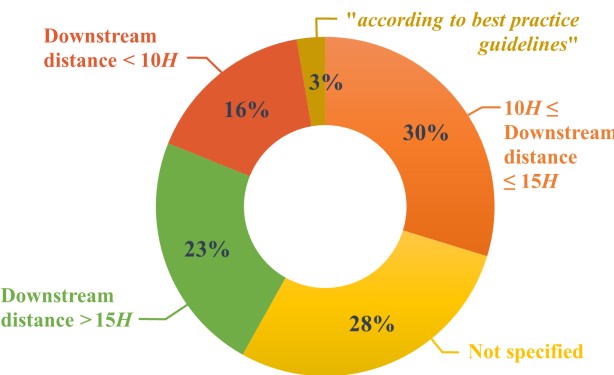

**Figure 10.** Downstream distance overview.

A concerning trend is observed: almost half of the reviewed publications either do not comply with the BPGs for the domain dimensions in the flow direction (e.g., [52,55–60], reaching zero distance offsets between the domain boundaries and the outer buildings in [61]), or do not report these important domain parameters (e.g., [16,62–64]). Conversely, there are articles [48,50,65–68] where the upstream distance that was used is much higher than the minimum recommendation of $5H$, ranging between $14H$ in [48] and $38H$ in [68]. The same is true for the downstream distance, which, in papers [15,48,50,68–71] ranges from $28H$ in [69] to $133.3H$ in [50].

### 3.4.3. Lateral Extension and Width of the Domain

There are two parameters that can describe the domain dimension in the lateral direction of the flow. These are the *lateral extension*, defined as the horizontal distance between the built area and the lateral boundaries of the computational domain, and the overall *domain width*, measuring the distance between the lateral boundaries themselves.

The best practice recommendations from [29,30] give prescriptions for the *lateral extension of the domain*. These are summarized in Table 5.

**Table 5.** Best practice guidelines on domain width.

| Publication Title | Year | Single Building | Multiple Buildings | Wind Tunnel Experiment | Real Terrain with Building Surroundings |
|---|---|---|---|---|---|
| Best practice guidelines for the CFD simulation of flow in the urban environment, quality assurance and improvement in microscale meteorological models (COST 732) [29] | 2007 | Calculated based on the height of the computational domain and the required blockage ($<3\%$) | Lateral extensions smaller than $5H$ can be used, where $H$ is the height of the *tallest* building. At least 2 different distances should be tested | Minimum of (the wind tunnel's test section width; built area width $+ 5H$ on either side of the geometry), where $H$ is the height of the *tallest* building | N/A |
| AIJ guidelines for practical applications of CFD to pedestrian wind environment around buildings [30] | 2008 | Minimum $5H$ | N/A | N/A | About $5H$ from the outer edges of the target building (maintaining a blockage ratio $\leq 3\%$) |

According to COST 732 recommendations, [29], the domain height should be selected first, and then, based on this choice, the lateral distance should be calculated, so that the blockage ratio is maintained below 3%. For a single building model with a domain height equal to $6H$, the minimum offset between the building and the lateral boundaries would

be $2.3 \times \max(H, W)$, where $H$ denotes the height, and $W$, the width of the building under consideration. Despite this general guideline based on the blockage ratio, COST 732 states that many authors recommend a much larger extension of at least $5H$, because the influence of the lateral boundaries on the flow and dispersion in the region of interest is highly case-dependent. It is also recommended to test at least two different distances from the built area. The only specific guideline regarding the domain dimensions that VDI 3783 [39] provides is maintaining a blockage ratio below 10%.

One more paper from this overview offers a recommendation for the domain width, [46], and points to $2.5H$ as a minimum safe value for this parameter. It should be noted that this analysis is based on an LES model of a street canyon, and therefore cannot be categorized in the general sections presented in Table 5. As in paper [72], most of the authors in the reviewed publications prefer to report the *total domain width*, instead of the *lateral domain extensions*; therefore, the summary presented in Figure 11 depicts the information provided for the *domain width* parameter.

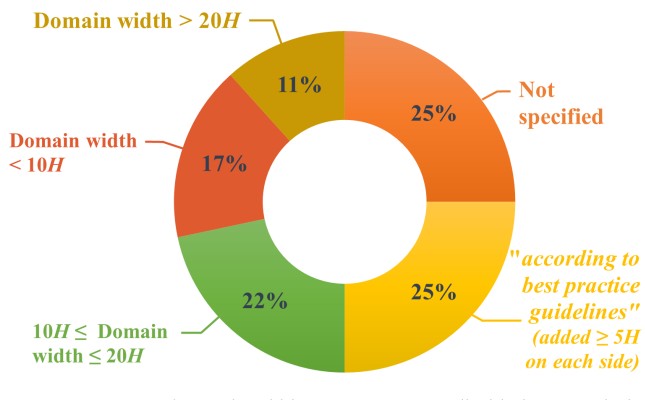

**Figure 11.** Domain width overview.

It should be noted that the lateral dimension of the domain is not relevant in two-dimensional analysis; therefore, publications that only utilized this type of analysis were filtered and excluded from this study. Following the process outlined in Section 3.1, the remaining 61 publications were divided into 5 categories. Similar to the trends observed for the other domain dimensions, over 40% of the authors either did not comply with the BPG (e.g., [52–55,69,73]), or did not report the domain width used in their models (e.g., [62,74–76]). Others, such as [16,48–50,77–79], selected much higher values for the investigated parameter, ranging between $24H$ in [77] and $333.3H$ in [50].

*3.5. Mesh*

Urban environments can reach high levels of geometry complexity at present, which leads to increased complexity in the computational grid required for analysis. Adequate discretisation is essential to CFD modelling, and this statement has been supported by many researchers, [29,32,80]. Generating a suitable mesh for complex urban geometries can take up the majority of the overall modelling time, but this effort can substantially decrease the computational time.

In terms of discretisation, two key factors determine the mesh quality. These are the *grid resolution* and the *cell quality*.

3.5.1. Grid Resolution

The *grid resolution* indicates whether the mesh elements are small enough to capture the investigated physical phenomena at the level of detail chosen for the study. This is associated with the minimum and maximum *cell sizes* and the respective *levels of refinement* where they are needed. This choice of mesh resolution should be very well balanced with respect to the required level of detail and the computational resources and time available

to the researcher. The authors of this study have access to statistics on mesh sizing and the maximum number of cells for the reviewed papers, but these will not be presented here. The reason for this is that these parameters are more or less dependent on other factors, such as software, urban environment type and scale, and mathematical models, and therefore cannot provide straightforward trends that the reader can easily interpret without considering the accompanying factors.

Other statistics, such as the number of grid cells resolving the obstacles, pollutant source, and the main areas of interest, are discussed in the best practice guidelines [29,30,39]. However, the majority of the reviewed publications do not report such parameters. AIJ [30] argues for a finer grid arrangement near the corners of the buildings to resolve the flow at which separation can occur. Within the vicinity of the target building, including the evaluation points' locations, they advise that the minimum grid resolution should be about 1/10 of the building scale (about 0.5 m to 5.0 m). There is another requirement regarding the evaluation height (usually, this is a pedestrian height between 1.5 m to 5.0 m above ground). It needs to be located at the third grid cell above the ground surface or higher. The recommendations of COST 732 [29] are similar. As an initial grid resolution, they suggest at least 10 cells per cube root of the building volume. The same is applicable to the vertical resolution of a street canyon with a width-to-height ratio of one. VDI guidelines [39] propose a less stringent approach in terms of building discretisation. They prescribe the buildings and areas of interest to be resolved with at least three grid points. However, relevant flow phenomena such as leeward vortices, street canyons, and flow reversion in the roof zone should be resolved with at least five (preferably eight) grid points per spatial direction. Furthermore, the surface boundary layer should be refined vertically to at least twice the height of the tallest relevant building, with at least 12 layers, with a maximum cell height of 10 m. Similarly to the AIJ prescriptions, at least three grid cells should be generated between the evaluation height and the ground surface.

### 3.5.2. Cell Type

The second factor that can be considered fundamental to the overall mesh quality is the *cell type*, along with the cell shape and distribution (refinement ratio, skewness, etc.). With respect to the calculation process in urban CFD simulations, the hexahedral cells (in 3D), and quadrilateral cells (in 2D) are considered to provide the most accurate results in terms of avoiding computational errors due to discretisation. At the opposite end of the spectrum are the pyramid and tetrahedral cells (in 3D), and triangles (in 2D). In cases where a purely hexahedral mesh is not attainable (for example, in complex geometries), the poly-hexcore (mosaic) meshing might be an invaluable compromise in terms of accuracy and computational time. This technology generates octree hex mesh in the bulk region, high-quality poly-prism mesh in the boundary layer, and ensures the conformal transition between the two with general polyhedral elements [81].

The comparison of the cell types used in the reviewed 2D studies are presented in Figure 12a, and the cell types for the 2.5D and 3D studies in Figure 12b.

In both cases, most papers followed the best practice approach and utilized a purely hexahedral mesh (or quadrilateral in 2D analysis). However, a concerning trend is observed, showing that over 30% of the publications do not specify the type of cells that they used in their study. The authors of this paper strongly believe that every paper that claims credibility should report the mesh type used in their work.

As prescribed in the best practice guidelines [29,30,39], the solution's independence from the selected grid should be confirmed with the verification procedures described in Section 3.9.1.

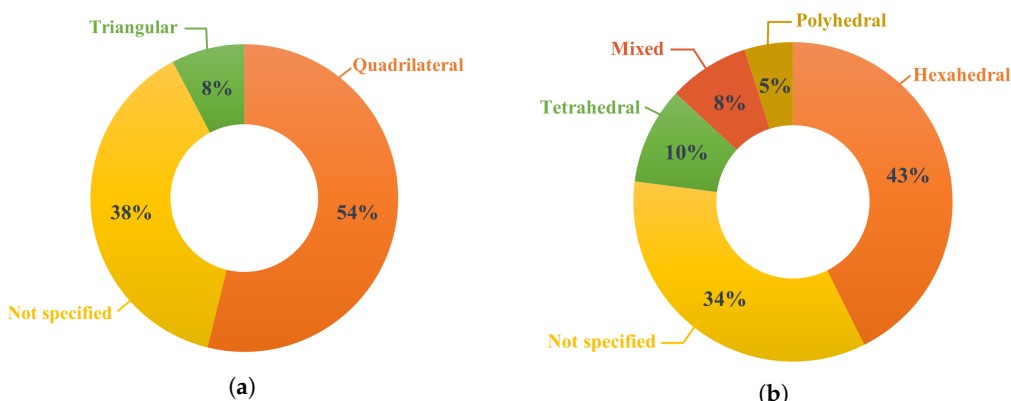

**Figure 12.** Cell types used in publications. (**a**) Cell types used in 2D analysis; (**b**) Cell types used in 3D analysis.

### 3.6. Physics

This section will provide an overview of the different CFD methods used in the literature to model the pollutant dispersion processes in the atmosphere. First, the use of thermal stratification will be addressed. Second, some of the pollutants' features—type of pollutant, reactivity, and source of emission—will be summarized. The governing equations that describe the spatial dispersion of the pollutants will then be investigated. Next, the use of steady-state and transient models and the implementation of different turbulence models will be explored. Finally, the wall treatment methods and the Turbulent Schmidt number values used in urban pollutant dispersion studies will be examined.

#### 3.6.1. Thermal Stratification

Nearly half of the examined articles do not explicitly state whether thermal energy equations are considered. Meanwhile, only five publications considered the thermal effects to better represent the pollutant dispersion in an urban environment. Both VDI and COST 732 guidelines state that neutral atmospheric stratification can be assumed in many cases, depending on its relevance to the investigated case [29,39]. AIJ does not provide any input on thermal stratification [30] .

#### 3.6.2. Pollutant Type

Out of the 74 reviewed articles, 53 investigated gaseous pollutants, such as carbon monoxide (CO), sulfur hexafluoride ($SF_6$), ethylene ($C_2H_4$), nitrogen oxides ($NO_x$), ethane ($C_2H_6$), etc. (e.g., [46,50,56,61,67,70,72,73,82,83]). A total of 13 papers modelled the pollutant as a particulate matter (PM), which included both PM10 and PM2.5 [48,74,77,84–93]. In addition, the pollutant in 3 studies consisted of both a gas and PM. Finally, 5 articles did not specify the type of pollutant used.

#### 3.6.3. Source of Emission

As vehicles play a significant role in urban pollution, the bulk of the reviewed cases employed a ground-level source of emission to mirror traffic pollution, as evidenced by Figure 13. Meanwhile, some articles (e.g., [58,59,68,71,73,75,94–96]) investigated the dispersion of pollutant from stack sources located on top of buildings, which is an equally important factor as it can negatively influence the air quality in the surrounding areas. In contrast, the pollutant in 3 studies [87,88,90] was injected into the computational domain from the inlet plane. Finally, a single study [97] explored the dispersion of a tracer gas released from the 3rd floor of a high-rise building. These statistics imply that the dispersion of traffic emissions is better investigated compared to the dispersion from other sources of pollution.

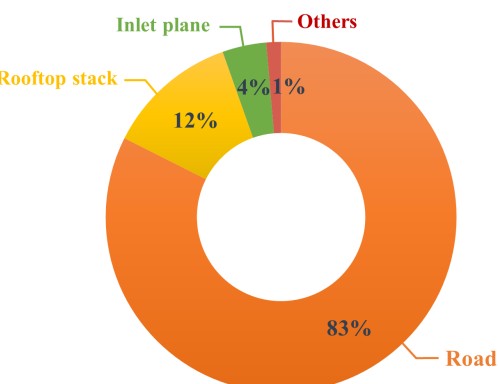

**Figure 13.** Types of emission sources in the investigated publications.

### 3.6.4. Pollutant Reactivity

Many articles do not explicitly comment on the chemical reactivity of the pollutant they used. However, all of those that state this used non-reactive gases. A study conducted by Santiago et al. [98] compared a reactive and non-reactive $NO_2$ pollutant within a real urban canopy. It concluded that, under winter conditions the modelling error was about 10%, which is acceptable in scenarios of pollutant dispersion, and the pollutant can be assumed to be non-reactive.

In terms of BPGs, none of the three reviewed documents, [29,30,39], provide recommendations regarding pollutant reactivity. AIJ's guidelines [30] fo not specifically addresss pollutant dispersion but rather the pedestrian wind environment around buildings, and COST action 732 is focused on urban wind flow and the dispersion of a passive scalar with similar density as the background fluid, where thermodynamic and chemical processes are not considered, [29]. Nevertheless, Section 5.3 of [29] mentions that physical complexities such as chemical reactions, break-up, coalescence, evaporation and particle–particle interactions might have to be modeled in some cases, without providing further specifications.

### 3.6.5. Governing Equations for Pollutant Dispersion

Two of the most commonly used methodologies for modelling the dispersion of pollutants in urban environment, as demonstrated in Figure 14, are the *passive scalar transport* and the *multi-species transport model*. If the former is used, the concentration of pollutant does not affect the flow field and its properties. If the multi-species transport model is used, however, the density of the pollutant is taken into consideration, which influences the flow field and its characteristics [99]. The PM motion, on the other hand, can be modeled by the Discrete Phase Model (DPM) or a revised generalized drift flux model. More information about these models can be found in [77,86–88,90]. Lastly, one study [82] used the mixture model to model the dispersion of pollutants. A total of 12 articles did not provide information on the governing equations they used in their studies.

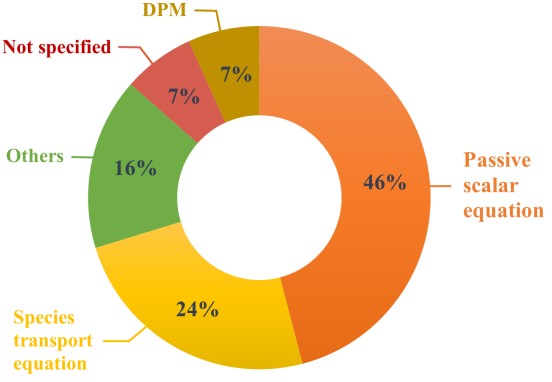

**Figure 14.** Types of governing equations for pollutant transport used in the investigated articles.

None of the reviewed BGPs documents, [29,30,39] explicitly specify the governing equations to be used in pollutant dispersion modelling.

### 3.6.6. Steady-State vs. Transient Models

Based on Figure 15, the majority of the studies employ the Reynolds-Averaged Navier Stokes (RANS) equations, which is a steady-state model. The rest of the reported simulations use unsteady models such as Large Eddy Simulation (LES), Unsteady RANS (URANS), Detached Eddy Simulation (DES) or a hybrid RANS-LES. Two studies compared different numerical approaches: [72] evaluated and compared the wind flow and pollutant dispersion in a street canyon using RANS and LES, while [100] investigated the performance of RANS, LES and DES in simulating the distribution of pollutant in building arrays. Both articles conclude that LES is the best option for predicting the mean velocity and mean pollutant concentration, but realize that it is more computationally expensive. Tominaga, et al. [24] support this statement but also declare that steady-RANS is acceptable in cases when advection effects are dominant. It is worth noting that, perhaps unsurprisingly, none of the reviewed papers utilise direct numerical simulation (DNS).

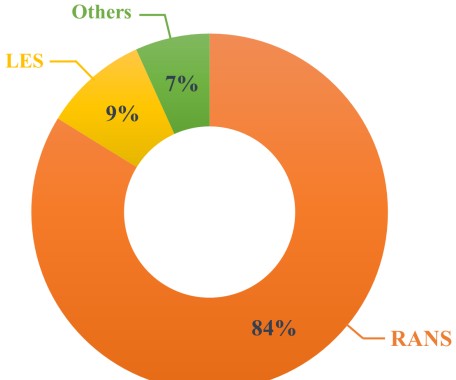

**Figure 15.** Types of simulations used in the examined publications.

The best practice guidelines in [29] review the application and limitations of RANS, URANS, LES and hybrid methods. Similar to [72,100], they conclude that LES and DES generally perform better than RANS and URANS, and could provide more information about the flow field and pollutant concentrations. However, the transient approaches require substantially greater computing times and are extremely sensitive to the inflow boundary conditions, requiring highly resolved experimental data (in terms of space and time) that are rarely available in practice. The AIJ working group does not give any specific prescriptions regarding the model time scale. They only point out that steady-state models cannot capture the unsteady periodic fluctuations that usually occur in urban landscapes but such approaches are sensitive to the turbulence models and boundary conditions used [30]. VDI Standard 3783 Part 9 [39] does not mention the selection of steady-state or transient models.

### 3.6.7. Turbulence Model

The analysis of the papers reviewed in the current work shows that, among the studies that employ RANS equations, as seen in Figure 16, the preferred turbulence model is the $k$–$\epsilon$, where the standard formulation is the most widely used, followed by the renormalization group (RNG) and the realizable $k$–$\epsilon$. Despite the popularity of the $k$–$\omega$ model for general CFD simulations and its better ability to resolve the boundary layers, "it suffers from the weakness of an extremely high sensitivity to $k$ and $\omega$ values at the inlet boundary" [70]. This could be a possible explanation of why it was only the model of choice in 3 articles. Further, two papers [101,102] propose an improved $k$–$\omega$ SST model due to Menter's $k$–$\omega$ SST inability to produce satisfactory results. Among the other turbulence models used in the reviewed articles are Durbin $k$–$\epsilon$, Murakami–Mochida–Kondo (MMK) $k$–$\epsilon$ and the

Reynolds Stress Model (RSM). After a thorough review, [28] concluded that the turbulence model should be chosen in accordance with its characteristics and limitations.

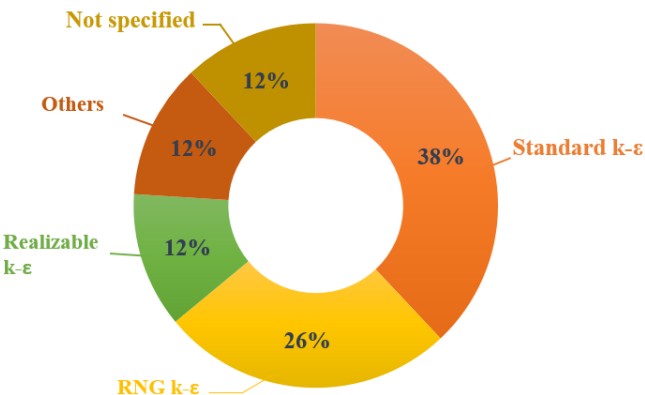

**Figure 16.** Types of turbulence models used in the investigated articles.

The best practice guidelines provided by the COST 732 working group discuss a number of turbulence models applicable to RANS, URANS, LES, and hybrid approaches, but refrain from giving recommendations regarding their usage, as the choice of an optimal turbulent model would be highly dependent on the specific application and the mesh resolution [29]. Rather, a validation strategy is proposed to assess the performance of the different turbulent models, along with examples on common practices in different organisations. AIJ, [30], generally agrees with the COST group's findings, and VDI, [39], also states that there is no universally optimal turbulence model.

### 3.6.8. Wall Treatment

Flow behaviour near a solid wall significantly differs from free-stream turbulent flows [103]. Consequently, a different set of equations is required for the accurate representation of the flow in the boundary layer. The inner turbulent boundary layer near a smooth solid surface consists of three main regions: viscous sub-layer ($y^+$ (in this work, $y^+$ is used to denote the non-dimensional distance from the geometry wall to the center of the mesh cell, as appropriate for the calculation) $< 5$); buffer layer ($5 < y^+ < 30$) and log-law layer ($30 < y^+ < 500$) [103]. There are different formulations for the wall treatment, which may require different $y^+$ values. Low $y^+$ wall treatments focus on resolving the viscous sub-layer, which requires a fine grid near the wall. Alternatively, wall functions can be used, which rely on the existence of the logarithmic region and are mainly applied for high-*Re*-number flows, where the viscous sub-layer is very thin [104]. Lastly, the all-$y^+$ treatment combines the other two methods to resolve the buffer region of the boundary layer.

Most of the investigated articles do not provide any information on the type of wall treatment used in their studies. Among those that do, however, wall functions are the preferred option, e.g., standard wall function (ANSYS Fluent), fully rough wall function (PHOENICS), nutkRoughWallFunction (OpenFOAM) (e.g., [53,84,86,105–107]). Few studies have used the all-$y^+$ wall treatment, of which the most popular is the enhanced wall treatment used in ANSYS Fluent (e.g., [100,108–110]). The use of wall functions requires fewer cells near the ground and building walls, and thus decreases the computational cost, which could explain why it is the preferred option in urban CFD simulations.

The BPGs given by COST 732 state that both low-$y^+$ and wall function approaches are valid [29]. Both COST 732, [29], and AIJ, [30], comment that the use of wall functions to represent flow around buildings is not entirely correct, but, due to separation points always being located at the leading edges at any *Re* numbers, the decrease in accuracy is insignificant. VDI [39] does not provide any insight on that matter.

### 3.6.9. Turbulent Schmidt Number

The turbulent Schmidt number is a free parameter, which occurs in the pollutant transport equation and represents the ratio between the rate of turbulent transport of momentum and the turbulent transport of mass [61]. Its value depends on the flow fields and the geometry, and it plays an important role in the accurate prediction of pollutant dispersion [61]. All the investigated papers used a value between 0.2 and 1.3. In addition, 3 articles [61,99,111] investigated the influence of the value of the turbulent Schmidt number on pollutant dispersion, and all of them concluded that the effects should be studied individually for each case in order to determine the most appropriate value. Ref. [28] also supports this statement and declares that the turbulent Schmidt number has a functional dependency on local flow properties. Finally, Ref. [61] suggests that different turbulence models could require different turbulent Schmidt number values.

None of the three reviewed best-practice guidelines documents [29,30,39] provide recommendations on defining the values of the turbulent Schmidt number.

### 3.7. Boundary Conditions

The boundary conditions (BC) of a CFD model should represent the urban environment and the physical processes that occur in it as closely as possible. In micro-scale wind-field modelling, only a small portion of the atmosphere located near the ground is investigated. This is usually enclosed in a box-shaped region, confined by six planes: one at the top, one at the bottom, and four at the sides, corresponding to the inflow, outflow, and the lateral boundaries of the domain. Any additional relevant obstacles and urban layout elements must be modelled explicitly and also assigned as boundaries. In most cases, only the bottom boundary (representing the ground surface) and the urban elements' boundaries (most often representing building walls and roofs) are physical boundaries. They should help to recreate the actual flow behaviour and, for that purpose, if necessary, could be assigned to a corresponding surface roughness. Unless the CFD model is an exact replica of a wind tunnel experiment, the rest of the boundaries are imaginary, and their function is to confine the model space and to apply the necessary environmental conditions without causing any non-physical disturbances in the flow.

### 3.7.1. Domain Air Inflow Boundary

The air inflow boundary condition is used for the application of the main wind flow characteristics (velocity profiles, turbulence properties, etc.), and to provide a realistic representation of the upstream roughness length. These characteristics can vary with time (for URANS, LES, DNS, or other unsteady simulations), or remain constant for steady-state RANS modelling. They can be taken from experimental or field data, from larger-scale models (meso-scale models), or from empirical, analytical, and theoretical expressions, or can be created using artificial stochastic data generation methods. The most commonly used expressions describing the vertical profiles of the flow properties for steady RANS simulations (mean velocity, turbulent kinetic energy, and turbulent dissipation rate) are discussed in [29,30,32]. The current publication takes a more general overview of the inflow boundary condition in terms of its type. None of the above-mentioned BPGs take such a perspective on the subject.

Figure 17 represents an overview of the air inflow boundary type in the reviewed publications. A clear trend is observed, that 80% of the authors applied the *Velocity inlet* boundary condition in their CFD models. In reality, this percentage could grow even further, since 16% of the publications do not report the type of inflow boundary condition they use.

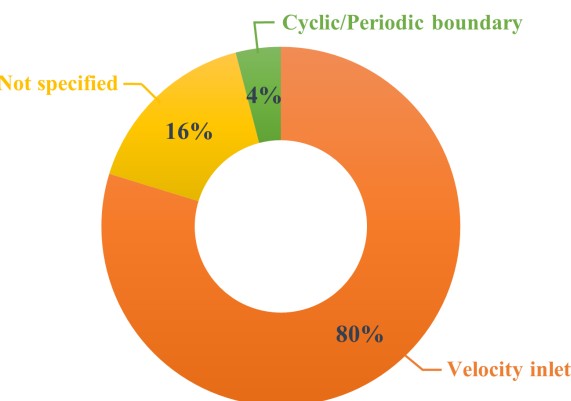

**Figure 17.** Inflow boundary type.

3.7.2. Domain Outflow Boundary

According to VDI [39]: *"At lateral outflow boundaries, reflection-free outward flow of disturbances generated in the model domain is achieved either through suitable specification of the variables themselves or through requirements placed on their spatial derivatives."* The COST Action 732 guidelines [29] go one step further and define four general types of outflow boundary conditions used in commercial CFD and microscale obstacle-accommodating meteorology models:

- *Outflow boundary*, corresponding to a fully developed flow where all flow derivatives are set to zero. Flow cannot re-enter the domain; this is the reason for the minimum downstream distances described in Section 3.4.
- *Pressure outlet*, with a constant static pressure and all other flow derivatives set to zero. Flow cannot re-enter the domain.
- *Radiation open boundary* used in microscale obstacle-accommodating meteorological models. Flow could re-enter the domain could.
- *Convective outflow boundary* that should be used in LES analysis.

On the other hand, AIJ [30] guidelines report *outflow* boundary condition, where the normal gradients of all variables are set to zero, to be the most commonly used.

The publications selected for this overview are sorted based on the boundary type they utilize in the region at which the fluid is exiting the domain. A summary of the results is presented in Figure 18.

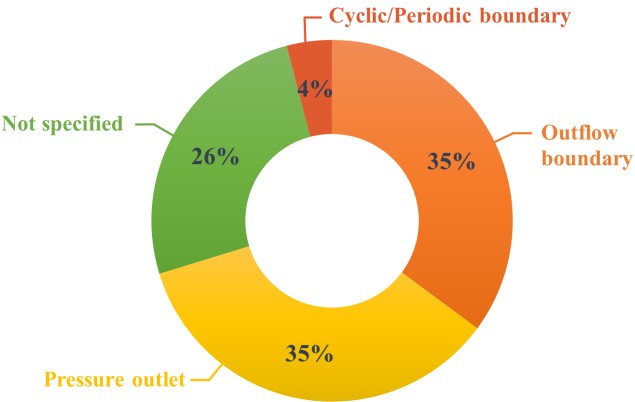

**Figure 18.** Outflow boundary type.

From the papers that explicitly state the boundary type that was used, it can be concluded that the *outflow* and the *pressure outlet* boundary types are the most common choices for urban CFD dispersion modelling, both with 35% representation. It is interesting to note that, in one of the studies, [50] instead of a zero constant pressure at the pressure outlet, the authors applied a pressure profile. More than a quarter of all publications

do not report the type of the outflow boundary used in their studies ([93,98,109,112] amongst others).

### 3.7.3. Domain Top Boundary

The general recommendation in the BPGs [29,30,39] is that the lateral and top surfaces of the computational domain do not considerably alter the wind flow characteristics and pollutant transport in the area of interest. This condition should be satisfied with the proper selection of a boundary condition type, along with respecting the minimum distances of the computational domain discussed in Section 3.4.

According to the AIJ guidelines [30], the top surface of the computational domain should be assigned an inviscid wall condition (normal velocity component and normal gradients of tangential velocity components set to zero) to make the computation more stable.

COST 732 [29] recommends that a constant shear stress or alternatively constant values for the velocities and the turbulence quantities are assigned to the top boundary, corresponding to the inflow profile at the respective height. The prescription of fixed values of the main variables is also the recommendation of [39]. The *free slip* condition is reported as being rarely used in microscale obstacle-accommodating meteorology models, and *symmetry* and *outflow* conditions are claimed to alter the inflow boundary profiles, and therefore should be avoided. In the case of the recreation of a wind tunnel experiment obtained in a closed test section, it is advised that the top boundary is modeled as a *solid wall* [29].

The reviewed publications' modelling pollution dispersion in an urban environment was sorted into different categories based on their choice of domain upper boundary type, and the results are presented in Figure 19.

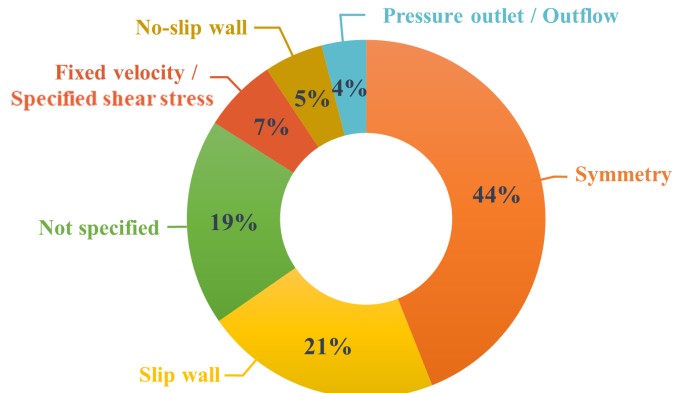

**Figure 19.** Top boundary type.

Most of the reviewed papers (44%) applied a *symmetry* constraint at the top boundary of the domain. This assumption is in contradiction with the COST 732 recommendations [29], which state that such an approximation can only be used if the top domain is outside the boundary layer. According to the latest measurements, the ABL average height ranges from a few hundred meters to a few kilometers, [113]; therefore, the assumption of a *symmetry* boundary condition would not be acceptable in most cases. Only 7% of the publications followed the best practice recommendation to apply fixed values either for the shear stress or for the velocity and turbulence parameters at the upper surface of the domain [79,86,89,92,101]. Nineteen percent did not specify the top boundary condition used. These negative trends and statistics raise the question of whether the equilibrium boundary layer profiles are sustained throughout the domain and also a question about the validity of the input flow data used in the majority of the studies.

### 3.7.4. Domain Lateral Boundaries

Similarly to the top domain boundary, the lateral surfaces parallel to the wind flow direction should not cause any artificial flow acceleration or non-physical phenomena.

COST 732 [29] reports that the *symmetry* condition is the most commonly used in commercial CFD codes but should be applied along with the recommended minimum distances discussed in Section 3.4. For the cases where computations are compared to the wind tunnel measurements performed within a closed test section, *solid walls* are reported as being a suitable boundary type.

Lateral domain boundaries parallel to the wind flow direction would only exist in the three-dimensional space. Therefore, the publications that investigated two-dimensional domains were excluded from this analysis. The remaining 61 papers were divided into categories based on their choice of a boundary type for the domain side walls, and the results are presented in Figure 20.

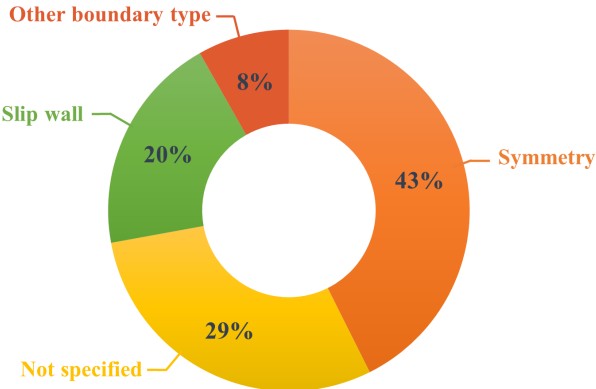

**Figure 20.** Lateral boundaries type.

By far, the largest group of reviewed papers, a full 43%, followed the best practice recommendation by applying a *symmetry* condition to the lateral boundaries of the model, and 20% used a *slip wall*. In the category summarized by the name: "Other boundary type", one study employed a *pressure outlet* [114], two papers used a *no-slip wall* [61,72], and two more studies used *periodic* boundary conditions [46,84]. Almost 30% of the papers did not specify the boundary type applied to the lateral domain walls, e.g., [51,65,70,109].

3.7.5. Domain Bottom Boundary

The bottom boundary of the domain normally represents the ground surface and/or urban ground elements and infrastructure. All reviewed best practice guidelines documents recommend verifying the horizontal homogeneity of the ABL in a domain without buildings before proceeding to the analysis of the actual model. The selected boundary condition for the bottom surface should be able to reproduce the gradual change in the vertical velocity profile that occurs near the ground as the flow proceeds downstream and retain its similarity to the wind tunnel measurement data, which should be confirmed at more than one measuring location. AIJ [30] suggests that such behaviour can be achieved using a logarithmic law variable distribution, either for a smooth wall or for a wall with roughness characteristics. Different possibilities for the log-law equations describing the boundary layer are also discussed. Both VDI [39] and COST 732 [29] recommend that, for the bottom physical edge of the domain, a *no-slip* condition is assumed, where the wind velocity components are set to zero. COST 732 guidelines further discuss the application, advantages, and disadvantages of different modelling approaches for *smooth* (low-Reynolds number approach and adoption of wall functions), and for *rough walls* (wall functions application and distributed roughness approach using a porous region).

In this overview, the publications did not follow a homogeneous approach to reporting the boundary conditions: 57% report using *no-slip wall* for the bottom domain boundary (e.g., [107,115–117]), 26% refer to the used condition simply as a *wall* (e.g., [51,75,118]), and the remaining 17% do not specify the type of boundary condition they used.

### 3.7.6. Domain Boundaries for Urban Elements (Buildings)

The domain physical boundaries that represent buildings and other urban elements acting as obstacles to the wind flow can be modelled explicitly, i.e., with their actual physical dimensions, or implicitly via a specified surface roughness. The guidelines provided for the ground surface by [29,30], and discussed in the previous section, hold true for the modelling of the building walls as well. The general recommendation is to apply a *no-slip boundary* condition to building walls and roofs with either *smooth* or *rough surfaces*. The majority of the reviewed publications investigating pollution dispersion in urban environments, 53%, apply the *no-slip boundary wall* to the building surfaces in their models (e.g., in papers [45,119–122]), another 26% report it simply as a *wall*, and the remaining 21% do not specify the boundary conditions that were utilized.

### 3.7.7. Boundary Conditions—Final Remarks

All of the guidelines presented in the previous sections regarding the domain boundary types applied in urban CFD dispersion modelling should mainly be considered mainly in their physical sense, i.e., to what extent the actual wind flow and environmental characteristics were successfully recreated in the model. The most appropriate boundary type, however, can vary for specific problems, and can also depend on the numerical methods and software selected for the analysis. Therefore, this choice must always be checked using V&V procedures and reported.

### 3.8. Solution Setup

The solution setup can consist of choosing appropriate discretisation schemes, convergence criteria, time steps (if the simulation is transient), etc. It can be as important to the accuracy of the final results as the mesh quality, boundary conditions and turbulence modelling. In [32], Blocken states that first-order discretisation schemes should be avoided at all cost due to their problems with numerical diffusion. COST 732 and AIJ guidelines [29,30] both agree that first-order schemes are not appropriate for convection and advection terms, unless used only for initial iterations. Thus, urban CFD simulations require at least second-order discretisation schemes for the final solution. None of the reviewed articles implemented first-order discretisation schemes, which is a sign of a well-conducted study. COST 732 also suggests that a double precision solver is ussed to decrease round-off errors, [29].

The convergence criteria in the investigated papers refer to scaled residuals for the main parameters of the governing equations, with the range $10^{-3}$ to $10^{-7}$. COST 732 guidelines recommend a reduction in the residuals of at least four orders of magnitude but also point out that a much lower limit should be used for the validation of turbulent processes, [29]. Furthermore, different urban configurations and different analysis software may require different criteria, which is why this study will not compare the convergence criteria in the reviewed articles. Nevertheless, the general advice by [29,30,32] holds true in all cases, and states that residuals should be allowed to reach a point at which they no longer change. In addition to the residuals, target variables, and integral balances of mass, momentum, and energy should also be monitored and oscillate around a constant value. The solution convergence can be considered to be achieved when all of the above conditions are satisfied.

### 3.9. A Note on Verification, Validation and Predictive Capability Estimation

The increasing importance of computer models as a substitute for experimentation was discussed in Section 1. When tackling problems related to air pollution dispersion, modelling plays a key role. This is because, often, physically investigating important scenarios may require significant resources from stakeholders, or have long-term detrimental effects for the health and well-being of the general population, making such experiments infeasible. However, models are only approximations of reality and are always built on assumptions. These assumptions may include, for example, incomplete knowledge about

the modelled processes and implementational idealisations. In order to present a reliable alternative to in situ experimentation, computer models must go beyond implementing complex integro-differential models and providing impactful visualisations. The fact that the reliability of predictions cannot be achieved by merely adding complexity to the model has long been recognised in many fields of science and engineering, and especially in modelling fluid flow [123–127]. Increasing model complexity only serves to mask the deficiencies of models, because each parameter carries additional uncertainty and increases the possibility of generating nonphysical predictions. The level of model reliability must instead be demonstrated by critically comparing predictions with experimental data. A considerable amount of confusion surrounds these activities, as there are those who claim that the verification and validation of atmospheric models are impossible on philosophical grounds [128]. To help clarify and establish a firm foundation for these concepts, several standards on predictive capability of models have been developed (see, e.g., [129–132]). Despite the fact that philosophical arguments cannot be considered an excuse to avoid practical engineering, such research seems to have established itself among wind engineers and continues to have a significant impact on the reach and quality of present day model reliability efforts, as discussed below. However, a key component that is missing from research refuting model reliability estimation is the careful and systematic application of uncertainty quantification (UQ) methods [133] (the field of UQ is not without its own controversies), which seeks to acknowledge and properly account for conditions that are unknown, but may be important.

This section examines each of the publications included in the review for their application of verification, validation and predictive capability estimation procedures in ensuring the published results are applicable to the investigated problems (in this work, for the sake of clarity, the abbreviation V&V is accepted to encompass the activities of verification, validation, predictive capability estimation and those parts of uncertainty quantification that are needed to support the former). The large body of literature on V&V [126,134,135] has been used as a benchmark against which to assess the steps taken to ensure predictive reliability results.

### 3.9.1. Verification

Verification is defined as the process of determining that a computer model implementation accurately represents the conceptual description of the model and the solution to that model [129]. In a more practical language, verification consists of steps to check the quality of the software (SQA) implementation, to assess the accuracy of the numerical algorithms comprising the model, and to estimate the uncertainty in results when the model is used for the generation of predictions. The former two activities comprise code verification, while the latter is formally known as solution verification [135]. Each of the 74 research articles reviewed were examined for a number of features regarding how they present different aspects of verification. Key features are summarised in Table 6. First, it became clear that none of the authors refer to verification work simply as *verification*. This is chiefly because none of the articles performed a complete verification campaign. Instead, efforts were focused on estimating spatial discretisation errors, which are only one of the five commonly accepted contributors to numerical inaccuracies [136] (see the second column of Table 6). It is interesting to note that, in 13 publications, no verification effort is reported, or one is not clearly described. Another point to observe is the use of the term *sensitivity analysis*, which authors use to denote studies aimed at determining the effect of various features in their models, despite the fact that sensitivity analysis has other, well-established uses in computational modelling (see e.g., [137]). Only 4 of the 61 articles describing some form of verification include temporal discretisation error estimation, and a single one discusses iterative convergence errors. There are also discussions of matters that are not formally considered part of verification (such as turbulence models, model-specific parameters and others). An important aspect of verification is the provision of quantitative evidence of the accuracy (or lack thereof) of the computer code. Despite this, only 9 of the

61 articles reporting on verification employ eda grid convergence index (GCI) [138], which is considered the standard when quantifying uncertainty due to spatial discretisation errors. The rest employ visual techniques to assess convergence. Two more papers claim to be using GCI, but do not calculate this in accordance with formal procedures. Most studies (33) constructed only two separate grids and often report "satisfying" results despite the lack of convergence in investigated quantities. An issue common to all reviewed articles is that none seem to appreciate the fact that spatial convergence must be investigated in the asymptotic grid region [139]. Another clear trend that emerges from the reviewed literature is that close to a quarter (11) of the papers that performed some verification presented their results as a comparison with experimental data and selected "converged" meshes as those closest to the data. This goes against V&V recommendations [126], which state that verification must be carried out prior to and separate from data-related activities. In addition, some articles report that no verification was performed because they claim to use a model similar to ones already verified in previously published research. This was found to be problematic either because the models in the two articles did not actually exhibit similarity in all relevant aspects, or because the article that was claimed to have performed verification had not done so. A final note that must be made is that none of the articles mention whether the authors are aware of any code verification activities. It is seldom the case that code verification is the provenance of engineers and urbanists, yet the responsibility lies with them to make sure the tools that they are using do what they claim to be doing.

**Table 6.** Features of verification. The name that different authors give to solution verification is given in the first column, in alphabetical order, except for the "Other", "Unclear" and "None" categories. Characteristics for all papers that have reported verification (61) are given in the second and third column.

| What It Is Called | | Errors Considered | | Verification with Data | | Code Verification | |
|---|---|---|---|---|---|---|---|
| Grid convergence | 2 | Spatial | 61 | Yes | 11 | Yes | 0 |
| Grid independence | 13 | Temporal | 4 | No | 50 | No | 74 |
| Grid sensitivity | 17 | Iterative | 1 | | | | |
| Grid study | 2 | Statistical | 0 | | | | |
| Grid refinement | 2 | Round-off | 0 | | | | |
| Mesh independence | 8 | Human | 0 | | | | |
| Mesh sensitivity | 2 | | | | | | |
| Sensitivity analysis | 6 | | | | | | |
| Other | 9 | Other | 2 | | | | |
| Unclear | 4 | | | | | | |
| None | 9 | | | | | | |

### 3.9.2. Validation

*Validation* is defined as the process of determining the degree to which a model is an accurate representation of the real world from the perspective of the intended uses of the model [131]. There are several resources with recommendations on what constitutes a good validation campaign, both in general [126,134,135,140] and in the case of modelling wind flow and pollution dispersion [31,127]. The three main points from these resources are:

- Validation data should be as relevant as possible to the cases in which the model predictions are required;
- Assessments of model accuracy must be quantitative, objective and independent of the decision on whether the model is good;
- This decision must be made in accordance with the model application.

Similar to the verification review, each article was examined against a number of criteria that ought to be present in high-quality validation campaigns. The results are summarised in Table 7. Much like in verification, validation does not seem to be a unanimously established term for the process of assessing model accuracy through experimental

data, despite the fact that it is the most common name. Some authors call the process "comparison" or even "verification". Some of the authors (11) that attempted data-related work were unclear on how to refer to the process. Two articles did not discuss validation.

**Table 7.** Features of validation. The name that different authors give to validation is given in the first column, in alphabetical order, except for the "Other", "Unclear" and "None" categories.

| What It Is Called | | Measure | | Accuracy Requirement | | Data Relevance | | Mix with Calibration | |
|---|---|---|---|---|---|---|---|---|---|
| Comparison | 11 | $d$ | 2 | On measure | 26 | Yes | 26 | Yes | 25 |
| Validation | 44 | FB | 21 | None | 48 | No | 48 | No | 49 |
| Verification | 2 | FAC2 | 19 | | | | | | |
| | | NMSE | 25 | | | | | | |
| | | $R^2$ | 21 | | | | | | |
| | | VG | 4 | | | | | | |
| | | Difference | 11 | | | | | | |
| Other | 4 | | | | | | | | |
| Unclear | 11 | | | | | | | | |
| None | 2 | None | 14 | | | | | | |

Since validation must present an objective assessment of the accuracy of the computer model, the techniques used to measure this accuracy should be equally objective and informative. However, all articles that included a validation step in their work resorted to some form of a summary measure that averages across different comparisons between data and prediction. Such measures include the index of agreement, $d$ [141], the fraction of predictions within a factor of two of observations (FAC2), the fractional bias (FB), the normalised mean squared error (NMSE) and the geometric variance (VG) [142] (this article also discusses the use of a bootstrap step to develop confidence measures regarding the results of the statistics. Only two of the reviewed articles employed some form of bootstrap bounding. Moreover, in general, bootstrap ought to be used with care). Various kinds of differences were also widely used in the reviewed papers for the assessment of model accuracy. Despite the fact that these are point-wise measures and, therefore, provide a fuller picture of the discrepancies between predictions and observations, none of the articles reported values across the distribution of measured quantities. A popular choice of (dis)agreement measures was the coefficient of determination (of linear fit between predictions and observations) and the correlation coefficient. The authors using these measures, however, made no comment on the common assumptions that they made, such as the fact that they measure linear stochastic relationship and their freely interpretable ranges of what constitutes a good value [143]. Finally, close to one fifth of the papers (14) used no quantitative means of assessing comparisons. Instead, they relied on a visual assessment, which is in direct violation of the objectivity requirement of formal validation. It is worth noting that all of the quantitative measures (except correlation coefficient and coefficient of determination) used in the reviewed literature come with some form of acceptable ranges. This feature is problematic, because it confounds the steps of validation and model adequacy, where the latter seeks to answer the question about whether the model, with the accuracy as computed in the validation, can be used for the intended application. Instead, acceptable ranges and other decision thresholds should be left to the consumers of the simulation results.

An important characteristic of a high-quality validation campaign is that the data used to assess model accuracy are relevant to the intended application of the model. This point is in close correspondence with the ideas outlined in the paragraph above, which emphasises the application-driven nature of validation. The urban geometry in the review literature varied widely, with some papers studying air pollution dispersion in settings as simple as symmetric, two-dimensional street canyons, while others investigated the same process over several city blocks represented in three dimensions. Despite this diversity, nearly 65% of the reviewed studies utilised data that were very far from the subsequent application. For instance, the model was to be used to simulate pollution dispersion over a real city

square, but the model was validated with pollution measurements in a flow over a set of regular cubes. This state of affairs shows that there is no clear appreciation of the utility of validation, where validation is seen more as a necessary step to satisfy academic journal reviewers, rather than a way to build trust in the model. Instead, validation data should be as close to the intended application as possible, so as to minimise the need to extrapolate the calculated accuracy figures.

The final validation trait investigated in the reviewed papers was the absence of a clear separation between validation and calibration. True validation mandates that the model remains unchanged during the process, which serves to obtain a clearer appreciation of its accuracy [144]. Despite the fact that the two activities are profoundly distinct, just over one third of the papers (25) employed some form of calibration during what they claimed to be the validation process. Among the adjusted model features were the turbulent Schmidt number and turbulence models. In all of these cases, the "validated" model was chosen as the one whose parametric value produced the closest match with the data.

### 3.9.3. Uncertainty Quantification and Predictive Capability

Another major issue related to the misuse of data is the fact that none of the reviewed publications take any uncertainty into account. Since most of the uncertainty incurred during validation is due to differences between the validation setup and the application setup, the lack of recognition of the necessity of uncertainty quantification is a natural cause for authors seeing no need to use the relevant data in their studies. Although eleven papers discuss measurement errors, these are not propagated through validation procedures or utilised in any other way. This is in violation of standards that clearly recognise the need for uncertainty-aware pollution dispersion and general wind models [27,145]. The lack of uncertainty quantification efforts, combined with the use of irrelevant data, preclude the estimation of any sensible predictive ability for the used models. This, in turn, renders these models as nothing more than generators of beautiful visualisations.

### 4. Conclusions and Future Work

The current paper aimed to summarize the trends in recent numerical studies, identify major challenges related to the CFD modelling of urban air pollution dispersion, and spark ideas for further exploration. Due to the high volume of research papers in the field, inclusion and exclusion criteria were applied to limit the extent and focus the scope of this review. Based on a selected group of 74 papers published in the period 2012–2022, information about over 190 model features was identified, extracted, and grouped into seven main categories: general information, geometry, mesh, physics, boundary conditions, solution setup, and V&V. Then, following a systematic approach, the information collected for each feature was analyzed and presented in a comprehensive manner, so that a qualitative and quantitative assessment, along with a comparison to the existing best practice guidelines (where applicable), could be made, revealing the trends in current modelling CFD techniques. Some of the key findings are summarized below:

- *Type of study*: Most of the reviewed papers (65%) were not related to immediate real-world applications but were in the research field of work.
- *Software usage*: However biased the reasons behind the software selection in the reviewed papers were, more than half of all studies used ANSYS Fluent software.
- *Geometry*: Urban environment type definition is too heterogeneous in nature to be assessed in a comprehensive manner, which points to the need for a unified and standardized classification in the field, to be used by all researchers. Such a classification would also facilitate the application of particular guidelines that are suitable for the different urban categories.
- *Computational domain*: In terms of computational domain size, more than half of the reviewed papers complied with the BPGs [29,30,39] or claimed to do so. However, close to 20% did not follow the established recommendations, which could be a prerequisite for acquiring non-physical results.

- *Mesh*: Cell type, shape, and distribution (refinement ratio, skewness, etc.) are considered fundamental to the overall quality of the computational mesh. The reviewed publications show that, for 3D analysis, the hexahedral cell type is most commonly reported (in 43% of the cases), and for 2D cases, the quadrilateral (54%) is mostly utilized. In both cases, however, approximately 40% of the authors did not report the cell type used in their studies.
- *Type of pollutant*: Additional research on particulate matter dispersion may be advisable, as over 70% of the reviewed articles only explored the dispersion of gaseous pollutants.
- *Source of emission*: A similar trend was observed in terms of emission source, where 83% of all studies investigated the dispersion of traffic emissions.
- *Governing equations for pollutant dispersion*: The passive scalar equation (in 46% of the papers) and the species transport model (in 24% of the papers) were the most commonly used techniques for modelling the pollution transport.
- *Steady-state vs. transient models*: The large majority of the studies (84%) employed the steady-state RANS equations, while 9% used LES and 7% used other models.
- *Turbulent Schmidt number*: The investigated papers used a wide range of values between 0.2 and 1.3. Many authors agree that the choice of the turbulent Schmidt number value is case-specific and different values must be tested to the find optimal one.
- *Boundary conditions*: The most widely used boundary condition types are as follows: velocity inlet for the inflow boundary, outflow and pressure outlet for the outflow boundary, symmetry planes for the top and lateral boundaries, and no-slip wall for the bottom boundary and the building geometries. In general, the papers followed the trends described in the BPGs [29,30,39] (where available) regarding the applicability of boundary conditions. However, there were certain cases where a boundary condition that was advised to be avoided by the BPGs [29,30,39] was the most commonly used in the reviewed articles. Such is the case with the symmetry condition used for the top domain boundary.
- *Verification, validation, and predictive capability*: The majority of publications claimed to perform V&V, but, in reality, their activities did not follow the established V&V guidance. Most papers reported the accuracy of their models using hedge words, such as "good", "accurate", "close" and so on, and did not employ any formal predictive capability estimation. This renders any subsequent results indicative at best.

Finally, a concerning trend was revealed where authors neglected to report important aspects regarding their simulations. In some cases, up to a third of the reviewed publications omitted essential details regarding the used parameters.

As a result of this work, the following questions arise for future consideration:

- Is the creation of a unified regulatory framework for CFD air pollution modelling a feasible task?
- What measures could be taken to raise awareness among researchers and practitioners regarding the proper application of V&V, and UQ activities?

Despite recent advancements in air pollution dispersion modelling using CFD, there are still grounds for necessary future work. Given the rising relevance of the air quality issue, a better understanding of the related processes is vital to dealing with the problem. Only studies that exhibit a profound knowledge in the field are aware of and follow existing best-practice guidelines, report on their modelling choices in detail, and present a high-quality V&V campaign can be considered as reliable evidence for urban policy-making.

**Author Contributions:** All authors contributed to the formulation of research goals and the development of the methodology. M.P. and R.M. took an active part in data curation. M.P., R.M. and P.O.H. performed all necessary formal analysis and investigation of the collected data. M.P. was the main contributor towards the visualizations presented in the paper. All authors were involved in the preparation and creation of the original draft. M.P., R.M. and P.O.H. reviewed and edited the manuscript. The project was mainly supervised by D.P.-A. and P.O.H. while the project administration activities were led by D.P.-A. Financial support for the project leading to this publication was acquired by D.P.-A. and P.O.H. All authors have read and agreed to the published version of the manuscript.

**Funding:** This work is part of the GATE Project supported by the Horizon 2020 WIDESPREAD-2018-2020 TEAMING Phase 2 Programme under Grant Agreement No. 857155, and by Operational Programme Science and Education for Smart Growth under Grant Agreement No. BG05M2OP001-1.003-0002-C01. P.O. Hristov is funded by the Bulgarian National Scientific Fund under National Scientific Program "Petar Beron i NIE", agreement no. KP-06-DB/3. The authors gratefully acknowledge the support of the Scientific Fund of Sofia University, under agreement No. 80-10-11/10.05.2022.

**Institutional Review Board Statement:** Not applicable.

**Informed Consent Statement:** Not applicable.

**Data Availability Statement:** The complete data presented in this study are available on request from the corresponding author.

**Conflicts of Interest:** The authors declare no conflict of interest.

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
