# Peer review of "Air Pollution Dispersion Modelling in Urban Environment Using CFD: A Systematic Review"

_atmosphere, doi:10.3390/atmos13101640_

Round 1

Reviewer 1 Report

In this paper, 74 papers published from 2012 to 2022 were analyzed, and more than 190 model features were extracted and divided into seven categories. Then, based on a systematic approach, the information collected for each feature is analyzed, and qualitative and quantitative assessments are performed according to existing best practice guidelines to reveal current trends in CFD modeling techniques.

While the content of the article is very abundant, help us understand and learn the application of CFD in air pollution modeling, the manuscript needs to be revised before accepted for publication. My detailed comments are as follows:

1. In the summary section, the background section (line 1 - line 7) takes up too much space. Simplify your language.

2. In Chapter 2.1, the questions that this paper aims to answer are not consistent with the reasons listed, and there is no strong causal relationship.

3. In Figure 1, the source of the retrieved article is indicated, but the authority of the retrieved article is not indicated, it is suggested to add this part.

4. Line 405, 566, 823, 858 the new paragraph has to be indented.

5. The conclusion is too long. For example, lines 1023-1030 can be deleted, lines 1031-1042 should be simplified.

6. Lines 1088-1101, don't put it in the conclusion, it's not a conclusion.

7. In the main text, the position of the image must be consistent with the description.

Reviewer 2 Report

The authors present a paper titled: "Air Pollution Dispersion Modelling in Urban Environment Using CFD – A Systematic Review". The paper is well structure and offers a full variety of approaches together with a comprehensive and exhaustive analysis of the air pollution CFD modellimg. Although not everything can be citted due to the large amount of papers, model types, etc., the paper present a very interesting description from many different aspects or angles. In pasrticular, the paper discusses percentages in function of type of CFD attending to the steady state (RANS) or unsteady state simulations (LES, others). The paper is ready to be published in opinion of this reviewer without changes.

Reviewer 3 Report

In this paper, the authors carry out a detailed, technical, systematic review of the use of CFD to model atmospheric pollution in urban environments. They identify 90 relevant peer reviewed research papers, evaluating the specific CFD setups and approaches against parameters identified in relevant standards. As such, the review will be essential reading for any future studies in which it is proposed to use CFD to model the dispersion of atmospheric pollutants. Readers will gain an overview of the typical parameters used and how these compare to industry standards.

In terms of the detailed evaluation of the published papers against technical standards, I cannot fault the approach: it is very detailed, well written and the results are presented concisely and consistently in the form of pie charts. There are very few grammatical or spelling mistakes.

However, whilst I understand the approach that the authors have taken, I was left wondering at the end about what all of this technical analysis and evaluation meant in terms of performance. I got no real sense of how these models performed in terms of predicting actual concentrations of pollutants in urban environments, which I think is important. The authors did cover methods of validation that the various papers had used, including standard statistical measures; however, whilst the number of papers that used each statistical measure was itemised, no actual results were presented, which I think would have been useful for the reader. The authors argued that whilst these standard statistical evaluation approaches come with acceptability criteria, it is up to users of the software to determine their own thresholds. Nevertheless, certainly in the analysis of Gaussian dispersion modelling results, acceptability criteria are regularly used and reported. Therefore, for the CFD studies reviewed in this paper, it would be useful if a summary was provided of the results of the CFD simulations against statistical analysis and acceptability criteria. Similarly, I would have appreciated some details of the nature of the validation process that took place in these papers, i.e. which pollutants, what type of monitoring data and what type of environment (for the latter, perhaps it would have been useful to provide a summary of the typical urban settings for these CFD simulations earlier in the manuscript, so that we had more context for typical applications in which CFD was employed). It might also have been relevant to make comparisons between CFD modelling and other approaches such as Gaussian, so that users with certain preferences for modelling approaches might make an informed choice on whether to explore alternatives.

Round 2

Reviewer 1 Report

This manuscript has been well revised. I have no further comments.

Reviewer 3 Report

I appreciate the detailed reply to my comments from the authors, and the clarifications that they provided, which I am happy to accept. It is good to see the inclusion of Figure 1, as this is very helpful in clarifying the overall structure of the paper.